statistics

categorical variables, contingency table, generalized linear modelling

**Author for correspondence:**
M. Papathomas
e-mail: m.papathomas@st-andrews.ac.uk

# On the correspondence of deviances and maximum-likelihood and interval estimates from log-linear to logistic regression modelling

## W. Jing and M. Papathomas

School of Mathematics and Statistics, University of St Andrews, The Observatory, Buchanan Gardens, St Andrews KY16 9LZ, UK

MP, 0000-0002-5897-695X

Consider a set of categorical variables $\mathcal{P}$ where at least one, denoted by $Y$, is binary. The log-linear model that describes the contingency table counts implies a logistic regression model, with outcome $Y$. Extending results from Christensen (1997, *Log-linear models and logistic regression*, 2nd edn. New York, NY, Springer), we prove that the maximum-likelihood estimates (MLE) of the logistic regression parameters equals the MLE for the corresponding log-linear model parameters, also considering the case where contingency table factors are not present in the corresponding logistic regression and some of the contingency table cells are collapsed together. We prove that, asymptotically, standard errors are also equal. These results demonstrate the extent to which inferences from the log-linear framework translate to inferences within the logistic regression framework, on the magnitude of main effects and interactions. Finally, we prove that the deviance of the log-linear model is equal to the deviance of the corresponding logistic regression, provided that no cell observations are collapsed together when one or more factors in $\mathcal{P}\backslash\{Y\}$ become obsolete. We illustrate the derived results with the analysis of a real dataset.

# 1. Introduction

Let $v = \{v_1, \ldots, v_n\}$ denote a set of observations, $\boldsymbol{\theta} = \{\theta_1, \ldots, \theta_n\}$ a set of parameters, and consider known or nuisance quantities $\boldsymbol{\phi} = \{\phi_1, \ldots, \phi_n\}$. Now, $v_i$, $i = 1, \ldots, n$, belongs to the exponential family of distributions if its probability function can be written as

$$f(v_i|\theta_i, \phi_i) = \exp\left\{\frac{w_i}{\phi_i}[v_i\theta_i - b(\theta_i)] + c(v_i, \phi_i)\right\},$$

where $w = \{w_1, \ldots, w_n\}$ are known weights, and $\phi_i$ is the dispersion or scale parameter. Regarding first-order moments, $\mu_i \equiv E(v_i) = b'(\theta_i)$. A generalized linear model relates $\boldsymbol{\mu} = \{\mu_1, \ldots, \mu_n\}$ to covariates by setting $\zeta(\boldsymbol{\mu}) = X_d \gamma$, where $\zeta$ denotes the link function, $X_d$ the covariate design matrix and $\gamma$ a vector of parameters. For a single $\mu_i$, we write $\zeta_i(\mu_i) = X_{d(i)} \gamma$, where $X_{d(i)}$ denotes the $i$th row of $X_d$, defining $\zeta$ as a vector function $\zeta \equiv \{\zeta_1, \ldots, \zeta_n\}$.

Let $\mathcal{P}$ denote a finite set of $P$ categorical variables. Observations from $\mathcal{P}$ can be arranged as counts in a $P$-way contingency table, with cell counts denoted by $n_i$, $i = 1, \ldots, n_{\text{ll}}$. The 'll' indicator alludes to a log-linear model. The counts follow a Poisson distribution with $E(n_i) = \mu_i$. A Poisson log-linear interaction model, $\log(\boldsymbol{\mu}) = X_{\text{ll}} \boldsymbol{\lambda}$, is a generalized linear model that relates the expected counts to $\mathcal{P}$.

From Christensen [1], there is an association between log-linear modelling and multinomial logistic regression. Consider categorical variables $X$, $Y$ and $Z$, with $J_X$, $J_Y$ and $J_Z$ levels, respectively. Let $j_X$, $j_Y$, $j_Z$ be integer indices that describe the level of $X$, $Y$ and $Z$. In a multinomial logistic regression with outcome $Y$, one typically models the log-odds of an observation at level $j_Y + 1$ relative to one at level $j_Y$, $\log(p_{j_Y+1}/p_{j_Y})$, $j_Y = 0, \ldots, J_Y - 1$. This can be viewed as equivalent to fitting a log-linear model as

$$\log \left( \frac{P(Y = j_Y + 1 | X, Z)}{P(Y = j_Y | X, Z)} \right) = \log \left( \frac{P(Y = j_Y + 1, X, Z)}{P(Y = j_Y, X, Z)} \right) = \log(\mu_{j_Y+1, j_X, j_Z}) - \log(\mu_{j_Y, j_X, j_Z}).$$

For more details, see [1, Section 4.6] where, in addition to the above approach, the alternative of constructing a multinomial model to model the log-odds of an observation at level $j_Y$, $j_Y = 1, \ldots, J_Y - 1$, relative to one at fixed level $J_Y$ is considered. In this manuscript, we focus on the association between log-linear modelling and binary logistic regression. Assume that the categorical variable $Y$ is binary. Then, a logistic regression can be fitted with $Y$ as the outcome, and all or some of the remaining $P - 1$ variables as covariates. We write, $\text{logit}(\boldsymbol{p}) = X_{\text{lt}} \boldsymbol{\beta}$, $\boldsymbol{p} = (p_1, \ldots, p_{n_{\text{lt}}})$, using the 'lt' indicator for the logistic model, denoting by $p_i$ the conditional probability that $Y = 1$ given covariates $X_{\text{lt}(i)}$, and by $\boldsymbol{\beta}$ the vector of model parameters.

From Agresti [2], when $\mathcal{P}$ contains a binary $Y$, a log-linear model $\log(\boldsymbol{\mu}) = X_{\text{ll}} \boldsymbol{\lambda}$ implies a specific logistic regression model with parameters $\boldsymbol{\beta}$ defined uniquely by $\boldsymbol{\lambda}$. As $Y$ is binary, $j_Y = 0, 1$. Consider the log-linear model

$$\log(\mu_{j_Y, j_X, j_Z}) = \lambda + \lambda_{j_X}^X + \lambda_{j_Y}^Y + \lambda_{j_Z}^Z + \lambda_{j_X, j_Y}^{XY} + \lambda_{j_X, j_Z}^{XZ} + \lambda_{j_Y, j_Z}^{YZ}, \tag{M1}$$

where the superscript denotes the main effect or interaction term. Similar to the derivation above, the corresponding logistic regression model for the conditional odds ratios for $Y$ is

$$\begin{aligned}
\log \left( \frac{P(Y = 1 | X, Z)}{P(Y = 0 | X, Z)} \right) &= \log \left( \frac{P(Y = 1, X, Z)}{P(Y = 0, X, Z)} \right) \\
&= \log(\mu_{j_Y=1, j_X, j_Z}) - \log(\mu_{j_Y=0, j_X, j_Z}) \\
&= \lambda_1^Y - \lambda_0^Y + \lambda_{j_X, 1}^{XY} - \lambda_{j_X, 0}^{XY} + \lambda_{1, j_Z}^{YZ} - \lambda_{0, j_Z}^{YZ}.
\end{aligned}$$

This is a logistic regression with parameters, $\boldsymbol{\beta} = (\beta, \beta_{j_X}^X, \beta_{j_Z}^Z)$, so that, $\beta = \lambda_1^Y - \lambda_0^Y$, $\beta_{j_X}^X = \lambda_{j_X, 1}^{XY} - \lambda_{j_X, 0}^{XY}$ and $\beta_{j_Z}^Z = \lambda_{1, j_Z}^{YZ} - \lambda_{0, j_Z}^{YZ}$. Identifiability corner point constraints set all elements in $\boldsymbol{\lambda}$ with a zero subscript equal to zero. Then, $\beta = \lambda_1^Y$, $\beta_{j_X}^X = \lambda_{j_X, 1}^{XY}$ and $\beta_{j_Z}^Z = \lambda_{1, j_Z}^{YZ}$. This scales in a straightforward manner to larger log-linear models. If a factor does not interact with $Y$ in the log-linear model, this factor disappears from the corresponding logistic regression. Without any loss of generality, and to simplify the analysis and notation, we henceforth assume corner point constraints.

Considering the log-odds implied by a logistic regression, more than one log-linear models provide the same structure. For example, the log-linear model, $\log(\mu_{j_Y j_X j_Z}) = \lambda + \lambda_{j_X}^X + \lambda_{j_Y}^Y + \lambda_{j_Z}^Z + \lambda_{j_X j_Y}^{XY} + \lambda_{j_Y j_Z}^{YZ}$, implies the same conditional log-odds structure for $Y$ as (M1). However, as shown in Christensen [3, Section 3.3.2] in conjunction with Christensen [1, Sections 11.1 and 12.4], the log-linear model that determines exactly the same logistic structure is the one that contains all possible interaction terms between the categorical factors in $\mathcal{P} \backslash \{Y\}$. Other log-linear models, even when they imply the same log-odds, impose additional constraints on the logistic structure. To avoid any confusion, the description of our results in this manuscript will expound that the considered log-linear model contains all possible interaction terms between the categorical factors in $\mathcal{P} \backslash \{Y\}$.

The relationship between $\boldsymbol{\beta}$ and $\boldsymbol{\lambda}$ can be described as $\boldsymbol{\beta} = T \boldsymbol{\lambda}$, where $T$ is an incidence matrix [4]. In the context of this manuscript, matrix $T$ has one row for each element of $\boldsymbol{\beta}$, and one column for each element of $\boldsymbol{\lambda}$. The elements of $T$ are zero, except in the case where the element of $\boldsymbol{\beta}$ is defined by the corresponding element of $\boldsymbol{\lambda}$. The number of rows of $T$ cannot be greater than the number of columns.

In Papathomas [5], the correspondence between the two modelling frameworks within the Bayesian framework was studied, deriving exact and asymptotic results. In this manuscript, we focus on the frequentist framework, and derive results on maximum-likelihood estimates (MLE), interval estimates and deviances. Christensen [1] offers a comprehensive account of log-linear and logistic regression modelling. In Christensen [1, ch. 11], results on the equivalence between MLE and confidence intervals were derived. We extend these results, by also considering the case where factors present in the contingency table and log-linear model are not present in the corresponding logistic regression model, and some of the contingency table cells are collapsed together. This case is not considered in [1,2] or, to the best of our knowledge, in any other published work. As stated in theorem 3.2, the MLE for the parameters of the logistic regression equals the MLE for the corresponding parameters of the log-linear model. Theorem 3.3 states that, asymptotically, standard errors for the logistic regression and corresponding log-linear model parameters are equal. Subsequently, Wald confidence intervals [2] are asymptotically equal.

For theorem 3.4, we stipulate that the logistic model is fitted to a dataset where no cell observations are collapsed together when one or more factors in $\mathcal{P}\backslash\{Y\}$ are not present in the logistic regression. Then, we prove that the deviance of the log-linear model equals the deviance of the corresponding logistic regression. Christensen [1, p. 371] refers to this equality, by considering a simple logistic regression with two parameters and showing that the likelihood ratio test statistic (LRTS) for the log-linear model equals the LRTS for the logistic regression. This is done by using the invariance of the MLE and the properties of the product-binomial sampling scheme [1, Section 2.6]. Christensen [1, p. 365] also shows that applying the logistic regression to a contingency table implies that the sampling scheme of the contingency table is product-binomial instead of multinomial. As these results are based on a logistic regression with two parameters, a general mathematical proof is required, provided in appendix A.

In §2 we provide additional notation and essential derivations for the log-linear and logistic regression model, then §3 contains the main contributions in this manuscript. In §4, the correspondence from a log-linear to a logistic regression model is illustrated using real data. We conclude with a discussion, where we also consider possible practical implications of our results.

# 2. Deviances and the information matrix

The deviance of a generalized linear model is crucial for assessing goodness of fit [6]. Let $\hat{\boldsymbol{\theta}}$ denote the MLE of $\boldsymbol{\theta}$. Let $L(\boldsymbol{\theta}_{\text{sat}}, v)$ and $L(\boldsymbol{\theta}_{\text{sim}}, v)$ denote the log-likelihood for the saturated model, and for a simpler model, respectively. The deviance is defined as

$$D(\hat{\boldsymbol{\theta}}, v) = -2[L(\hat{\boldsymbol{\theta}}_{\text{sim}}, v) - L(\hat{\boldsymbol{\theta}}_{\text{sat}}, v)].$$

Then,

$$D(\hat{\boldsymbol{\theta}}, v) = -2\left(\sum_{i=1}^{n} \frac{w_i}{\phi_i}(v_i\hat{\theta}_{i,\text{sim}} - b(\hat{\theta}_{i,\text{sim}})) + c(v_i, \phi_i)\right.$$

$$\left. - \sum_{i=1}^{n} \frac{w_i}{\phi_i}(v_i\hat{\theta}_{i,\text{sat}} - b(\hat{\theta}_{i,\text{sat}})) + c(v_i, \phi_i)\right)$$

$$= -2\left(\sum_{i=1}^{n} \frac{w_i}{\phi_i}v_i(\hat{\theta}_{i,\text{sim}} - \hat{\theta}_{i,\text{sat}}) - \frac{w_i}{\phi_i}(b(\hat{\theta}_{i,\text{sim}}) - b(\hat{\theta}_{i,\text{sat}}))\right).$$

Denote by $\hat{\boldsymbol{\gamma}}$ the MLE of $\boldsymbol{\gamma}$, and $\mathcal{I}(\hat{\boldsymbol{\gamma}})$ the information matrix $X_d^\top \mathcal{V} X_d$. ($\mathcal{V}$ will be specified below for both modelling frameworks as $\mathcal{V}_{\text{log-linear}}$ and $\mathcal{V}_{\text{logistic}}$.) Then, from Agresti [2], asymptotically

$$\hat{\boldsymbol{\gamma}} \sim N(\boldsymbol{\gamma}, \mathcal{I}^{-1}).$$

## 2.1. Log-linear regression

Consider a vector $\boldsymbol{n}$ of counts $n_i$ $i = 1, \dots, n_{\text{ll}}$. Now, $N = \sum_{i=1}^{n_{\text{ll}}} n_i$, and,

$$f(n_i | \mu_i) = \frac{e^{-\mu_i}\mu_i^{n_i}}{n_i!},$$

with $\theta_i = \log(\mu_i)$, $b(\theta_i) = e^{\theta_i}$ and $c(n_i, \phi_i) = -\log(n_i!)$. Also, $w_i\phi_i^{-1} = 1$, so that $w_i = 1$ implies $\phi_i = 1$. Note that, $\mu_i = b'(\theta_i) = e^{\theta_i}$, and $\text{Var}(n_i) = \phi_i w_i^{-1} b''(\theta) = e^{\theta_i}$. For the log-linear model, $\log(\boldsymbol{\mu}) = X_{ll}\boldsymbol{\lambda}$, $X_{ll}$ is a $n_{ll} \times n_\lambda$ design matrix of covariates, and $\zeta_i(\mu_i) = \log(\mu_i)$. Given the above,

$$D(\hat{\boldsymbol{\mu}}, \boldsymbol{n}) = -2\left(\sum_{i=1}^{n_{ll}} n_i(\log(\hat{\mu}_i) - \log(n_i)) - \hat{\mu}_i + n_i\right)$$

$$= 2\sum_{i=1}^{n_{ll}} n_i\log\left(\frac{n_i}{\hat{\mu}_i}\right) - 2\sum_{i=1}^{n_{ll}} n_i + 2\sum_{i=1}^{n_{ll}} \hat{\mu}_i.$$

From Agresti [2, p. 140], when the log-linear model contains an intercept, $\sum_{i=1}^{n_{ll}} n_i = \sum_{i=1}^{n_{ll}} \hat{\mu}_i$. Then,

$$D(\hat{\boldsymbol{\mu}}, \boldsymbol{n}) = 2\sum_{i=1}^{n_{ll}} n_i\log\left(\frac{n_i}{\hat{\mu}_i}\right). \tag{2.1}$$

The diagonal matrix $\mathcal{V}_{\log-\text{linear}}$ has non-zero elements $\exp\{X_{ll(i)}\hat{\boldsymbol{\lambda}}\}$, $i = 1, \ldots, n_{ll}$.

## 2.2. Logistic regression

Assume that $y_i$, $i = 1, \ldots, n_{lt}$, is the proportion of successes out of $t_i$ trials. Now, $N = \sum_{i=1}^{n_{lt}} t_i$, and,

$$f(t_iy_i|p_i) = \binom{t_i}{t_iy_i} p_i^{t_iy_i}(1 - p_i)^{t_i - t_iy_i},$$

where $\theta_i = \text{logit}(p_i)$, $b(\theta_i) = \log(1 + e^{\theta_i})$ and $c(y_i, \phi_i) = \log\binom{t_i}{t_iy_i}$. Also, $w_i\phi_i^{-1} = t_i$, so that $w_i = 1$ implies $\phi_i = t_i^{-1}$. Note that

$$E(y_i) = b'(\theta_i) = \frac{e^{\theta_i}}{1 + e^{\theta_i}} = p_i \quad \text{and} \quad \text{Var}(y_i) = \frac{\phi_i}{w_i}b''(\theta_i) = \frac{1}{t_i}\frac{e^{\theta_i}}{(1 + e^{\theta_i})^2} = \frac{p_i(1 - p_i)}{t_i}.$$

For the logistic regression, $\text{logit}(\boldsymbol{p}) = X_{lt}\boldsymbol{\beta}$, $X_{lt}$ is a $n_{lt} \times n_\beta$ design matrix, and $\zeta_i(p_i) = \text{logit}(p_i)$. Given the above

$$D(\boldsymbol{y}, \hat{\boldsymbol{p}}) = -2\left(\sum_{i=1}^{n_{lt}} t_iy_i\left[\log\left(\frac{\hat{p}_i}{1 - \hat{p}_i}\right) - \log\left(\frac{y_i}{1 - y_i}\right)\right] - t_i\log\left(\frac{1}{1 - \hat{p}_i}\right) + t_i\log\left(\frac{1}{1 - y_i}\right)\right)$$

$$= -2\left(\sum_{i=1}^{n_{lt}} t_iy_i\log(\hat{p}_i) - t_iy_i\log(y_i)\right.$$

$$\left. + \sum_{i=1}^{n_{lt}} (t_i - t_iy_i)\log(1 - \hat{p}_i) - (t_i - t_iy_i)\log(1 - y_i)\right).$$

After some algebra,

$$D(\boldsymbol{y}, \hat{\boldsymbol{p}}) = 2\sum_{i=1}^{n_{lt}} t_iy_i\log\left(\frac{t_iy_i}{t_i\hat{p}_i}\right) + 2\sum_{i=1}^{n_{lt}} (t_i - t_iy_i)\log\left(\frac{t_i - t_iy_i}{t_i - t_i\hat{p}_i}\right)$$

$$= 2\sum_{i=1}^{n_{lt}} t_iy_i\log\left(\frac{y_i}{\hat{p}_i}\right) + 2\sum_{i=1}^{n_{lt}} (t_i - t_iy_i)\log\left(\frac{1 - y_i}{1 - \hat{p}_i}\right). \tag{2.2}$$

The diagonal matrix $\mathcal{V}_{\text{logistic}}$ has non-zero elements $t_i\exp\{X_{lt(i)}\hat{\boldsymbol{\beta}}\}\exp\{1 + X_{lt(i)}\hat{\boldsymbol{\beta}}\}^{-2}$, $i = 1, \ldots, n_{lt}$.

## 3. Results

To facilitate the derivation of theoretical results, we introduce the following additional notation. Without any loss of generality, let $x_{\cdot 1}$ be the binary $Y$ factor, and $x_{\cdot 2}, \ldots, x_{\cdot q}$ the $q - 1$ factors that are present in the log-linear model but disappear from the logistic regression model as they do not interact with $Y$. Denote the rest of the factors by $x_{\cdot q+1}, \ldots, x_{\cdot P}$. Each element of $\boldsymbol{n}$ is denoted by $n_j$, $\boldsymbol{j} = (j_1, \ldots, j_P)$, $0 \leq j_p \leq J_p - 1$, $p = 1, \ldots, P$, where $J_p$ is the number of levels of $x_{\cdot p}$. Here, $\boldsymbol{j}$, identifies the combination of variable levels that cross-classify the given cell. We define $L$ as the set of all $n_{ll}$ cross-classifications, so that, $L = \otimes_{p=1}^{P}[j_p]$. Elements $y_j$ and $\mu_j$ are defined analogously.

**Lemma 3.1.** *Assume that the log-linear model contains all possible interaction terms between the categorical factors in $\mathcal{P}\backslash\{Y\}$. Then, for all $0 \leq j_p \leq J_p - 1$, $p = 2, \ldots, P$,*

$$n_{0,j_2,\ldots,j_P} + n_{1,j_2,\ldots,j_P} = \hat{\mu}_{0,j_2,\ldots,j_P} + \hat{\mu}_{1,j_2,\ldots,j_P}.$$

*Proof.* The proof is given in appendix A. ∎

**Theorem 3.2.** *Assume that the log-linear model contains all possible interaction terms between the categorical factors in $\mathcal{P}\backslash\{Y\}$. Then, the MLE $\hat{\boldsymbol{\beta}}$ of the parameters of the logistic-regression is equal to the MLE of the corresponding parameters of the log-linear model.*

*Proof.* The proof is given in appendix A. ∎

**Theorem 3.3.** *Assume that the log-linear model contains all possible interaction terms between the categorical factors in $\mathcal{P}\backslash\{Y\}$. Then, asymptotically, the standard error for each element of $\boldsymbol{\beta}$ is equal to the standard error for the corresponding parameter of the log-linear model.*

*Proof.* The proof is given in appendix A. ∎

The proofs for theorems 3.2 and 3.3 include the case where factors present in the log-linear model are not present in the corresponding logistic regression and some of the contingency table cells are collapsed together. For completeness, our proofs also include the case where all factors in $\mathcal{P}\backslash\{Y\}$ are present in the logistic regression model. Theorem 3.4 postulates that $n_{lt} = n_{ll}/2$, i.e. the number of proportions fitted by the logistic regression should be half the number of cell counts in the contingency table. This happens either because all factors in $\mathcal{P}\backslash\{Y\}$ are present in the logistic regression, or because counts in cells with the same cross-classification considering $x_{.q+1}, \ldots, x_{.P}$ are not collapsed. This is important for observing equal deviances for the log-linear model and the corresponding logistic regression. Intuitively, when $n_{lt} = n_{ll}/2$, the number of observations fitted by the logistic regression is in direct correspondence with the number of observations fitted by the log-linear model. When $n_{lt} < n_{ll}/2$, a logistic regression model with the same number of parameters fits a smaller number of observations, something that naturally results in a smaller deviance compared to the deviance observed when the contingency table is not collapsed. This is illustrated in §4 with the analysis of a real dataset.

**Theorem 3.4.** *Assume that the log-linear model contains all possible interaction terms between the categorical factors in $\mathcal{P}\backslash\{Y\}$. Assume also that the corresponding logistic regression is fitted to a dataset where $n_{lt} = n_{ll}/2$. Then, the deviance of the log-linear model equals the deviance of the corresponding logistic regression.*

*Proof.* The proof is given in appendix A. ∎

## 4. Illustration

Edwards & Havránek [7] presented a $2^6$ contingency table in which 1841 men were cross-classified by six binary risk factors $\{A, B, C, D, E, F\}$ for coronary heart disease. Adopting the notation in Agresti [2], a single letter denotes the presence of a main effect, two-letter terms denote the presence of the implied first-order interaction and so on and so forth. The presence of an interaction between a set of variables implies the presence of all lower-order interactions plus main effects for that set. Consider the log-linear model

$$\log(\boldsymbol{\mu}) = AC + AD + AE + BCDEF. \tag{M2}$$

Treating $A$ as the outcome, the corresponding logistic regression is

$$\mathrm{logit}(\boldsymbol{p}) = C + D + E. \tag{M3}$$

The deviances, MLE and standard errors for the relevant parameters of both models are given in table 1, after fitting the models in R using the 'glm' function. We observe that corresponding quantities are equal. To obtain equal deviances, although factors $B$ and $F$ are not present in the logistic regression, the logistic model was fitted to a dataset where contingency table cell counts discriminated only by $B$ and $F$ were not collapsed together. This resulted in $n_{lt} = 32$. The datasets for (M2) and (M3) are given in appendix A. The design matrix $X_{lt}^{(M3)}$ is shown below, with $\top$ denoting the transpose, with some of

**Table 1.** Deviances, MLE and standard errors for the relevant parameters of log-linear model (M2) and the corresponding logistic regressions (M3) and (M4). (Standard errors are given in brackets.)

| log-linear model (M2), $\log(\boldsymbol{\mu}) = AC + AD + AE + BCDEF$, deviance = 33.51 | | | |
| --- | --- | --- | --- |
| | A | AC | AD | AE |
| MLE | −0.4140 (0.0892) | 0.5501 (0.0958) | −0.3684 (0.0967) | 0.4893 (0.0973) |
| outcome is A (M3), $\text{logit}(\boldsymbol{p}) = C + D + E$, deviance = 33.51 | | | |
| | intercept | C | D | E |
| MLE | −0.4140 (0.0892) | 0.5501 (0.0958) | −0.3684 (0.0967) | 0.4893 (0.0973) |
| outcome is A (M4), $\text{logit}(\boldsymbol{p}) = C + D + E$, deviance = 3.47 | | | |
| | intercept | C | D | E |
| MLE | −0.4140 (0.0892) | 0.5501 (0.0958) | −0.3684 (0.0967) | 0.4893 (0.0973) |

the rows identical.

$$
X_{\text{lt}}^{(M3)} = \begin{pmatrix}
1 & 1 & 1 & 1 & 1 & 1 & 1 & 1 & 1 & 1 & 1 & 1 & 1 & 1 & 1 & 1 & 1 & 1 & 1 & 1 \\
0 & 0 & 1 & 1 & 0 & 0 & 1 & 1 & 0 & 0 & 1 & 1 & 0 & 0 & 1 & 1 & 0 & 0 & 1 & 1 \\
0 & 0 & 0 & 0 & 1 & 1 & 1 & 1 & 0 & 0 & 0 & 0 & 1 & 1 & 1 & 1 & 0 & 0 & 0 & 0 \\
0 & 0 & 0 & 0 & 0 & 0 & 0 & 0 & 1 & 1 & 1 & 1 & 1 & 1 & 1 & 1 & 0 & 0 & 0 & 0
\end{pmatrix}
$$

$$
\begin{pmatrix}
1 & 1 & 1 & 1 & 1 & 1 & 1 & 1 & 1 & 1 & 1 & 1 \\
0 & 0 & 1 & 1 & 0 & 0 & 1 & 1 & 0 & 0 & 1 & 1 \\
1 & 1 & 1 & 1 & 0 & 0 & 0 & 0 & 1 & 1 & 1 & 1 \\
0 & 0 & 0 & 0 & 1 & 1 & 1 & 1 & 1 & 1 & 1 & 1
\end{pmatrix}^{\top} .
$$

As factors $B$ and $F$ disappear from the logistic regression that corresponds to (M2), one may decide to collapse together the contingency table cells with the same cross-classification considering $C$, $D$ and $E$. A logistic regression is fitted, denoted by (M4). It only contains main effects for $C$, $D$ and $E$, as does (M3). The dataset for (M4) is shown in appendix A. The design matrix for (M4) is

$$
X_{\text{lt}}^{(M4)} = \begin{pmatrix}
1 & 1 & 1 & 1 & 1 & 1 & 1 & 1 \\
0 & 1 & 0 & 1 & 0 & 1 & 0 & 1 \\
0 & 0 & 1 & 1 & 0 & 0 & 1 & 1 \\
0 & 0 & 0 & 0 & 1 & 1 & 1 & 1
\end{pmatrix}^{\top} .
$$

Relevant output is given in table 1. MLE and standard errors are equal, as theorems 3.2 and 3.3 hold. However, as cells are collapsed together and $n_{\text{lt}} \neq n_{\text{ll}}/2$, the deviances differ.

## 5. Discussion

The results in Christensen [1] and this manuscript demonstrate the extent to which inferences from the log-linear framework translate to inferences within the logistic regression framework, on the magnitude of main effects and interactions.

When factors are not present in the logistic regression, one may choose to collapse the counts in the contingency table cells that are only discriminated by the obsolete variables $x_{\cdot 2}, \ldots, x_{\cdot q}$. Logistic regression parameter estimates and associated standard errors are not affected by collapsing the cell counts. This is shown in the proofs for theorems 3.2 and 3.3 in appendix A. However, the logistic regression fitted to the collapsed dataset, returns a different deviance compared to a logistic regression with the same covariates (parameters) fitted without collapsing. This is expected, as two models with the same number of parameters are fitted to a different number of data points. The deviance naturally increases for the larger dataset.

Our results concern two of the most popular approaches for the analysis of categorical observations and the correspondence between them. Theoretical derivations on such associations improve understanding and enhance the models' use, as advances for one framework are not always readily available to the other. For instance, to describe the joint probability distribution between covariates, Zhou et al. [8] adopt a PARAFAC factorization. Marginal independence is modelled with fixed baseline vectors, providing expressions for parameters of the log-linear models that correspond to the

adopted latent class model. Another example is Papathomas & Richardson [9], where the use of employing variable selection within clustering to assist log-linear modelling is investigated, without examining logistic regression models.

Data accessibility. All data considered in this manuscript are provided in appendix A. This article has no additional data.

Authors' contributions. W.J. and M.P. contributed equally to all parts of this manuscript. All authors gave final approval for publication.

Competing interests. We declare we have no competing interests.

Funding. The first author acknowledge the support of the School of Mathematics and Statistics, as well as CREEM, at the University of St Andrews, and the University of St Andrews St Leonard's 7th Century Scholarship.

Acknowledgements. We are grateful to Prof. Ronald Christensen for the instructive discussions we had during the preparation of this manuscript. We are also grateful for the comments by two reviewers and the editor that improved this manuscript.

# Appendix A.

*Proof of Lemma 3.1.* To facilitate this and subsequent proofs, the following notation is introduced, similar to Papathomas [5]. Using the incidence matrix $T$ discussed in §1, write the mapping between $\beta$ and $\lambda$ as $\beta = T\lambda$, where

$$T = \begin{pmatrix} \lambda_{(1)} \\ \vdots \\ \lambda_{(n_{\lambda_Y})} \end{pmatrix},$$

and $\lambda_{(k)}$, $k = 1, \ldots, n_{\lambda_Y}$, is a vector of zeros with the exception of one element that is equal to one. This element is in the position of the $k$th $\lambda$ parameter with a $Y$ in its superscript. With $n_{\lambda_Y}$ we denote the number of parameters in $\lambda$ with a $Y$ in their superscript. To ease algebraic calculations, and without any loss of generality, rearrange the elements of $\lambda$, creating a new vector $\lambda_r$, so that $T$ changes accordingly to, $T_r = (I\ 0)$, where $I$ is an $n_\beta \times n_\beta$ identity matrix. (Vector $\mu$ is similarly rearranged to $\mu_r$.) The rows and columns of $X_{ll}$ are also rearranged accordingly to create $X_{rll}$, so that

$$X_{rll} = \begin{pmatrix} X_{lt}^* & X_{ll-lt} \\ 0 & X_{ll-lt} \end{pmatrix}. \tag{A1}$$

$X_{ll-lt}$ is a square $(n_{ll}/2 \times n_{ll}/2)$ matrix. This is because we consider the log-linear model that, in addition to the terms that involve $Y$, contains all possible interaction terms between the categorical factors in $\mathcal{P}\backslash\{Y\}$. The number of parameters that correspond to the intercept, main effects and interactions for $\mathcal{P}\backslash\{Y\}$ is $n_{ll}/2$. $X_{lt}^*$ is a $n_{ll}/2 \times n_\beta$ matrix. When $q = 1$, all factors other than $Y$ remain in the logistic regression model as covariates. When no cell counts are collapsed, either because $q = 1$, or because we opt not to collapse, $X_{lt}^* = X_{lt}$, and $n_{ll} = 2 \times n_{lt}$. When the cell counts that are only discriminated by the obsolete variables $x_{.2}, \ldots, x_{.q}$ are collapsed, by rearranging the rows of $X_{rll}$ when necessary, we can write $X_{lt}^*$ as, $X_{lt}^* = (X_{lt}^\top X_{lt}^\top \ldots X_{lt}^\top)^\top$, where $X_{lt}^\top$ is repeated $(J_1 - 1) \times J_2 \times \ldots \times J_q$ times. For example, for $q = 2$, $X_{lt}$ repeats $J_2$ times within $X_{lt}^*$, and $n_{ll} = 2 \times J_2 \times n_{lt}$. When $q = P$, the corresponding logistic regression model only contains an intercept, and one may decide to fit the logistic regression to a collapsed contingency table that only contains two cells describing the total number of counts where $Y = 0$ and $Y = 1$. Then, $n_{ll} = 2 \times J_2 \times \ldots \times J_P \times n_{lt}$.

We can now write $\beta = T_r \lambda_r$. For example, assume the log-linear model (M1) describes a $3 \times 2 \times 2$ contingency table. Then, $q = 1$, and the standard arrangement of the elements of $\lambda$ would be such that,

$$X_{ll} = \begin{pmatrix} 1 & 0 & 0 & 0 & 0 & 0 & 0 & 0 & 0 & 0 \\ 1 & 1 & 0 & 0 & 0 & 0 & 0 & 0 & 0 & 0 \\ 1 & 0 & 1 & 0 & 0 & 0 & 0 & 0 & 0 & 0 \\ 1 & 0 & 0 & 1 & 0 & 0 & 0 & 0 & 0 & 0 \\ 1 & 1 & 0 & 1 & 0 & 1 & 0 & 0 & 0 & 0 \\ 1 & 0 & 1 & 1 & 0 & 0 & 1 & 0 & 0 & 0 \\ 1 & 0 & 0 & 0 & 1 & 0 & 0 & 0 & 0 & 0 \\ 1 & 1 & 0 & 0 & 1 & 0 & 0 & 1 & 0 & 0 \\ 1 & 0 & 1 & 0 & 1 & 0 & 0 & 0 & 1 & 0 \\ 1 & 0 & 0 & 1 & 1 & 0 & 0 & 0 & 0 & 1 \\ 1 & 1 & 0 & 1 & 1 & 1 & 0 & 1 & 0 & 1 \\ 1 & 0 & 1 & 1 & 1 & 0 & 1 & 0 & 1 & 1 \end{pmatrix}, \quad \lambda = \begin{pmatrix} \lambda \\ \lambda_1^X \\ \lambda_2^X \\ \lambda_1^Y \\ \lambda_1^Z \\ \lambda_{11}^{XY} \\ \lambda_{21}^{XY} \\ \lambda_{11}^{XZ} \\ \lambda_{21}^{XZ} \\ \lambda_{11}^{YZ} \end{pmatrix}, \quad T = \begin{pmatrix} 0 & 0 & 0 & 1 & 0 & 0 & 0 & 0 & 0 & 0 \\ 0 & 0 & 0 & 0 & 0 & 1 & 0 & 0 & 0 & 0 \\ 0 & 0 & 0 & 0 & 0 & 0 & 1 & 0 & 0 & 0 \\ 0 & 0 & 0 & 0 & 0 & 0 & 0 & 0 & 0 & 1 \end{pmatrix}.$$

After rearranging

$$
X_{\mathrm{rll}} =
\begin{pmatrix}
1 & 0 & 0 & 0 & 1 & 0 & 0 & 0 & 0 & 0 \\
1 & 1 & 0 & 0 & 1 & 1 & 0 & 0 & 0 & 0 \\
1 & 0 & 1 & 0 & 1 & 0 & 1 & 0 & 0 & 0 \\
1 & 0 & 0 & 1 & 1 & 0 & 0 & 1 & 0 & 0 \\
1 & 1 & 0 & 1 & 1 & 1 & 0 & 1 & 1 & 0 \\
1 & 0 & 1 & 1 & 1 & 0 & 1 & 1 & 0 & 1 \\
0 & 0 & 0 & 0 & 1 & 0 & 0 & 0 & 0 & 0 \\
0 & 0 & 0 & 0 & 1 & 1 & 0 & 0 & 0 & 0 \\
0 & 0 & 0 & 0 & 1 & 0 & 1 & 0 & 0 & 0 \\
0 & 0 & 0 & 0 & 1 & 0 & 0 & 1 & 0 & 0 \\
0 & 0 & 0 & 0 & 1 & 1 & 0 & 1 & 1 & 0 \\
0 & 0 & 0 & 0 & 1 & 0 & 1 & 1 & 0 & 1
\end{pmatrix}, \quad
\lambda_r =
\begin{pmatrix}
\lambda_1^Y \\
\lambda_{11}^{XY} \\
\lambda_{21}^{XY} \\
\lambda_{11}^{YZ} \\
\lambda \\
\lambda_1^X \\
\lambda_2^X \\
\lambda_1^Z \\
\lambda_{11}^{XZ} \\
\lambda_{21}^{XZ}
\end{pmatrix}, \quad
T_r =
\begin{pmatrix}
1 & 0 & 0 & 0 & 0 & 0 & 0 & 0 & 0 & 0 \\
0 & 1 & 0 & 0 & 0 & 0 & 0 & 0 & 0 & 0 \\
0 & 0 & 1 & 0 & 0 & 0 & 0 & 0 & 0 & 0 \\
0 & 0 & 0 & 1 & 0 & 0 & 0 & 0 & 0 & 0
\end{pmatrix}.
$$

See Papathomas [5] for another example where $q = 2$. From Agresti [2, p. 138], the likelihood equations for a log-linear model $\log(\mu_r) = X_{\mathrm{rll}}\lambda_r$ are

$$
\sum_{j_1,\ldots,j_P} n_{j_1,\ldots,j_P} X_{\mathrm{rll}(j_1,\ldots,j_P),j} - \sum_{j_1,\ldots,j_P} \hat{\mu}_{j_1,\ldots,j_P} X_{\mathrm{rll}(j_1,\ldots,j_P),j} = 0,
$$

where $X_{\mathrm{rll}(j_1,\ldots,j_P),j}$ is the element of $X_{\mathrm{rll}}$ in the row that corresponds to $n_{j_1,\ldots,j_P}$, and column $j$, $j = 1, \ldots, n_\lambda$. As $\log(\mu_r) = X_{\mathrm{rll}}\lambda_r$, includes all interactions between factors other than $Y$, $X_{\mathrm{ll}-lt}$ is the design matrix for a saturated log-linear model for all factors other than $Y$. Because $X_{\mathrm{ll}-lt}$ repeats within $X_{\mathrm{rll}}$ (as shown in (A1)), the $n_{\mathrm{ll}}/2$ likelihood equations for $\log(\mu_r) = X_{\mathrm{rll}}\lambda_r$, $j = n_\beta + 1, \ldots, n_\lambda$, are also the likelihood equations of a saturated log-linear model for fitting the $n_{\mathrm{ll}}/2$ observations, $n_{0,j_2,\ldots,j_P} + n_{1,j_2,\ldots,j_P}$:

$$
\sum_{j_2,\ldots,j_P} (n_{0,j_2,\ldots,j_P} + n_{1,j_2,\ldots,j_P}) X_{\mathrm{ll}-lt(j_2,\ldots,j_P),j}
$$
$$
= \sum_{j_2,\ldots,j_P} (\hat{\mu}_{0,j_2,\ldots,j_P} + \hat{\mu}_{1,j_2,\ldots,j_P}) X_{\mathrm{ll}-lt(j_2,\ldots,j_P),j}.
$$

Here, $X_{\mathrm{ll}-lt(j_2,\ldots,j_P),j}$ is the element of $X_{\mathrm{ll}-lt}$ in the row that corresponds to $y_{j_2,\ldots,j_P}$, and column $j$, $j = n_\beta + 1, \ldots, n_\lambda$. As these are the likelihood equations of a saturated model,

$$
n_{0,j_2,\ldots,j_P} + n_{1,j_2,\ldots,j_P} = \hat{\mu}_{0,j_2,\ldots,j_P} + \hat{\mu}_{1,j_2,\ldots,j_P},
$$

and this completes the proof. ∎

*Proof of Theorem 3.1. All factors in $\mathcal{P}\backslash\{Y\}$ are present in the logistic regression, or no collapsing of cells.* From Agresti [2, (p.193)] the likelihood equations for the logistic regression model, $\mathrm{logit}(p) = X_{lt}\beta$, are

$$
\sum_{j_2,\ldots,j_P} t_{j_2,\ldots,j_P} y_{j_2,\ldots,j_P} X_{lt(j_2,\ldots,j_P),j} - \sum_{j_2,\ldots,j_P} t_{j_2,\ldots,j_P} \hat{p}_{1,j_2,\ldots,j_P} X_{lt(j_2,\ldots,j_P),j} = 0,
$$

for $j = 1, \ldots, n_\beta$. Now,

$$
\frac{\hat{\mu}_{1,j_2,\ldots,j_P}}{\hat{\mu}_{0,j_2,\ldots,j_P}} = \frac{\exp(X_{\mathrm{rll}(1,j_2,\ldots,j_P)}\hat{\lambda}_r)}{\exp(X_{\mathrm{rll}(0,j_2,\ldots,j_P)}[n_\beta + 1 : n_\lambda]\hat{\lambda}_r[n_\beta + 1 : n_\lambda])}
$$
$$
= \exp(X_{\mathrm{rll}(1,j_2,\ldots,j_P)}[1 : n_\beta]\hat{\lambda}_r[1 : n_\beta]) = \exp(X_{lt(j_2,\ldots,j_P)}\hat{\beta}) = \frac{\hat{p}_{1,j_2,\ldots,j_P}}{1 - \hat{p}_{1,j_2,\ldots,j_P}},
$$

where, $a[a_1 : a_2]$, specifies the vector formed by all elements from the $a_1$th to the $a_2$th element of vector $a$, including the $a_1$th and $a_2$th elements. Therefore,

$$
\hat{p}_{1,j_2,\ldots,j_P} = \frac{\hat{\mu}_{1,j_2,\ldots,j_P}}{\hat{\mu}_{0,j_2,\ldots,j_P} + \hat{\mu}_{1,j_2,\ldots,j_P}}.
$$

Thus, to estimate $\boldsymbol{\beta}$, the likelihood equations are

$$\sum_{j_2,\ldots,j_P} t_{j_2,\ldots,j_P} y_{j_2,\ldots,j_P} X_{\mathrm{lt}(j_2,\ldots,j_P),j}$$

$$- \sum_{j_2,\ldots,j_P} t_{j_2,\ldots,j_P} \frac{\hat{\mu}_{1,j_2,\ldots,j_P}}{\hat{\mu}_{0,j_2,\ldots,j_P} + \hat{\mu}_{1,j_2,\ldots,j_P}} X_{\mathrm{lt}(j_2,\ldots,j_P),j} = 0$$

$$\Rightarrow \sum_{j_2,\ldots,j_P} t_{j_2,\ldots,j_P} y_{j_2,\ldots,j_P} X_{\mathrm{lt}(j_2,\ldots,j_P),j} - \sum_{j_2,\ldots,j_P} \hat{\mu}_{1,j_2,\ldots,j_P} X_{\mathrm{lt}(j_2,\ldots,j_P),j} = 0.$$

For the log-linear model, for $\lambda_r[1:n_\beta]$, the likelihood equations are

$$\sum_{j_1,\ldots,j_P} n_{j_1,j_2,\ldots,j_P} X_{\mathrm{rll}(j_1,\ldots,j_P),j} - \sum_{j_1,\ldots,j_P} \hat{\mu}_{j_1,\ldots,j_P} X_{\mathrm{rll}(j_1,\ldots,j_P),j} = 0,$$

where $j = 1, \ldots, n_\beta$. As, $X_{\mathrm{rll}(0,j_2,\ldots,j_P),j} = 0$ for all $j$, the likelihood equations for estimating $\lambda_r[1:n_\beta]$ are

$$\sum_{j_2,\ldots,j_P} n_{1,j_2,\ldots,j_P} X_{\mathrm{rll}(1,j_2,\ldots,j_P),j} - \sum_{j_2,\ldots,j_P} \hat{\mu}_{1,j_2,\ldots,j_P} X_{\mathrm{rll}(1,j_2,\ldots,j_P),j} = 0.$$

As, $n_{1,j_2,\ldots,j_P} = t_{j_2,\ldots,j_P} \times y_{j_2,\ldots,j_P}$, and $X_{\mathrm{lt}(j_2,\ldots,j_P),j} = X_{\mathrm{rll}(1,j_2,\ldots,j_P),j}$, the likelihood equations for estimating $\boldsymbol{\beta}$ and the corresponding $\lambda_r[1:n_\beta]$ are the same. Therefore, $\hat{\boldsymbol{\beta}} = \hat{\lambda}_r[1:n_\beta]$, as the number of equations equals the number of parameters.

*Factors not present in the logistic regression, with collapsing of cells.* As $X_{\mathrm{lt}}$ repeats $J_2 \times \cdots \times J_q$ times within $X_{\mathrm{lt}}^*$, the likelihood equations for estimating $\lambda_r[1:n_\beta]$, for $j = 1, \ldots, n_\beta$, are shown below:

$$\sum_{j_2,\ldots,j_q} \sum_{j_{q+1},\ldots,j_P} n_{1,j_2,\ldots,j_P} X_{\mathrm{rll}(1,j_2,\ldots,j_P),j} - \sum_{j_2,\ldots,j_q} \sum_{j_{q+1},\ldots,j_P} \hat{\mu}_{1,j_2,\ldots,j_P} X_{\mathrm{rll}(1,j_2,\ldots,j_P),j} = 0,$$

$$\Rightarrow \sum_{j_{q+1},\ldots,j_P} t_{+_2,\ldots,+_q,j_{q+1},\ldots,j_P} y_{+_2,\ldots,+_q,j_{q+1},\ldots,j_P} X_{\mathrm{rll}(j_{q+1},\ldots,j_P),j}$$

$$- \sum_{j_{q+1},\ldots,j_P} \hat{\mu}_{1,+_2,\ldots,+_q,j_{q+1},\ldots,j_P} X_{\mathrm{rll}(j_{q+1},\ldots,j_P),j},$$

where

$$\hat{\mu}_{1,+_2,\ldots,+_q,j_{q+1}\ldots j_P} = \sum_{j_2,\ldots j_q} \hat{\mu}_{1,j_2,\ldots j_q,j_{q+1},\ldots j_P},$$

$$t_{+_2,\ldots,+_q,j_{q+1},\ldots j_P} = \sum_{j_2,\ldots j_q} t_{j_2,\ldots j_q,j_{q+1},\ldots j_P}$$

and

$$y_{+_2,\ldots,+_q,j_{q+1},\ldots j_P} = \sum_{j_2,\ldots j_q} y_{j_2,\ldots j_q,j_{q+1},\ldots j_P}.$$

These are also the equations for estimating the logistic regression parameters $\boldsymbol{\beta}$. So, $\hat{\boldsymbol{\beta}} = \hat{\lambda}_r[1:n_\beta]$, as the number of equations equals the number of parameters. ∎

*Proof of Theorem 3.2.* Consider a vector of cell counts $n = \{n_1, \ldots, n_{\mathrm{ll}}\}$, and the log-linear model $\log(\boldsymbol{\mu}) = X_{\mathrm{ll}}\boldsymbol{\lambda}$. Then, from Agresti [2], asymptotically:

$$\mathrm{Var}(\hat{\boldsymbol{\lambda}}) \simeq [\mathcal{I}(\hat{\boldsymbol{\lambda}})]^{-1} = [X_{\mathrm{ll}}^\top \mathcal{V}(\hat{\boldsymbol{\lambda}}) X_{\mathrm{ll}}]^{-1}.$$

After rearranging the rows and columns of $X_{\mathrm{ll}}$, consider the log-linear model with linear predictor $X_{\mathrm{rll}}\lambda_r$, for cell counts $n_r$, where $n_r$ is $n$ rearranged to correspond to $X_{\mathrm{rll}}$. Now

$$\mathrm{Var}(\hat{\boldsymbol{\lambda}}_r) \simeq [\mathcal{I}(\hat{\boldsymbol{\lambda}}_r)]^{-1} = [X_{\mathrm{rll}}^\top \mathcal{V}(\hat{\boldsymbol{\lambda}}_r) X_{\mathrm{rll}}]^{-1} = [X_{\mathrm{rll}}^\top (\mathcal{V}(\hat{\boldsymbol{\lambda}}_r)) X_{\mathrm{rll}}]^{-1}$$

$$= \left[ \left( \begin{pmatrix} \mathcal{V}_1\mathcal{V}_2 & \mathbf{0} \\ \mathbf{0} & \mathcal{V}_2 \end{pmatrix}^{1/2} \begin{pmatrix} X_{\mathrm{lt}}^* & X_{\mathrm{ll-lt}} \\ \mathbf{0} & X_{\mathrm{ll-lt}} \end{pmatrix} \right)^\top \right.$$

$$\left. \times \begin{pmatrix} \mathcal{V}_1\mathcal{V}_2 & \mathbf{0} \\ \mathbf{0} & \mathcal{V}_2 \end{pmatrix}^{1/2} \begin{pmatrix} X_{\mathrm{lt}}^* & X_{\mathrm{ll-lt}} \\ \mathbf{0} & X_{\mathrm{ll-lt}} \end{pmatrix} \right]^{-1}.$$

$\mathcal{V}_1$ denotes a diagonal matrix with non-zero elements $\exp(X_{\mathrm{lt}(i)}^*(\boldsymbol{T}_r\hat{\boldsymbol{\lambda}}_r))$, $i = 1, \ldots, n_{\mathrm{ll}}/2$. $\mathcal{V}_2$ denotes a diagonal matrix with non-zero elements $\exp(X_{\mathrm{ll-lt}(i)}\hat{\boldsymbol{\lambda}}_{\mathrm{ll-lt}})$, $i = 1, \ldots, n_{\mathrm{ll}}/2$, where $\hat{\boldsymbol{\lambda}}_{\mathrm{ll-lt}}$ denotes the MLE

for $\boldsymbol{\lambda}_r \backslash T_r \boldsymbol{\lambda}_r$. Now,

$$\mathrm{Var}(\hat{\boldsymbol{\lambda}}_r) \simeq \begin{pmatrix} X_{\mathrm{lt}}^{*\top} A_{12} X_{\mathrm{lt}}^* & X_{\mathrm{lt}}^{*\top} A_{12} X_{\mathrm{ll-lt}} \\ X_{\mathrm{ll-lt}}^{\top} A_{12} X_{\mathrm{lt}}^* & X_{\mathrm{ll-lt}}^{\top} (A_{12} + A_2) X_{\mathrm{ll-lt}} \end{pmatrix}^{-1},$$

where $A_{12} = \mathcal{V}_1 \mathcal{V}_2$ and $A_2 = \mathcal{V}_2$. From Lutkepohl [10, p. 147, result 2(a)], and Lutkepohl [10, p. 29, line 6], the submatrix $H$ that is formed by the first $n_\beta$ rows and columns of $\mathrm{Var}(\hat{\boldsymbol{\lambda}}_r)$ is

$$
\begin{aligned}
H &= [X_{\mathrm{lt}}^{*\top} A_{12} X_{\mathrm{lt}}^* - X_{\mathrm{lt}}^{*\top} A_{12} X_{\mathrm{ll-lt}} (X_{\mathrm{ll-lt}}^{\top}(A_{12}+A_2)X_{\mathrm{ll-lt}})^{-1} X_{\mathrm{ll-lt}}^{\top} A_{12} X_{\mathrm{lt}}^*]^{-1} \\
&= [X_{\mathrm{lt}}^{*\top} A_{12} X_{\mathrm{lt}}^* - X_{\mathrm{lt}}^{*\top} A_{12} X_{\mathrm{ll-lt}} X_{\mathrm{ll-lt}}^{-1}(A_{12}+A_2)^{-1}(X_{\mathrm{ll-lt}}^{\top})^{-1} X_{\mathrm{ll-lt}}^{\top} A_{12} X_{\mathrm{lt}}^*]^{-1} \\
&= [X_{\mathrm{lt}}^{*\top} A_{12} X_{\mathrm{lt}}^* - X_{\mathrm{lt}}^{*\top} A_{12}(A_{12}+A_2)^{-1} A_{12} X_{\mathrm{lt}}^*]^{-1} \\
&= [X_{\mathrm{lt}}^{*\top}(A_{12} - A_{12}(A_{12}+A_2)^{-1}A_{12})X_{\mathrm{lt}}^*]^{-1} \\
&= [X_{\mathrm{lt}}^{*\top}(A_{12} - A_{12}(A_{12}(I + A_{12}^{-1}A_2))^{-1}A_{12})X_{\mathrm{lt}}^*]^{-1} \\
&= [X_{\mathrm{lt}}^{*\top}(A_{12} - A_{12}(I + A_{12}^{-1}A_2)^{-1})X_{\mathrm{lt}}^*]^{-1}.
\end{aligned}
$$

Thus,

$$
\begin{aligned}
H &= [X_{\mathrm{lt}}^{*\top}(\mathcal{V}_1\mathcal{V}_2 - \mathcal{V}_1\mathcal{V}_2(I + \mathcal{V}_1^{-1}\mathcal{V}_2^{-1}\mathcal{V}_2)^{-1})X_{\mathrm{lt}}^*]^{-1} \\
&= [X_{\mathrm{lt}}^{*\top}(\mathcal{V}_1\mathcal{V}_2 - \mathcal{V}_1^2\mathcal{V}_2(I + \mathcal{V}_1)^{-1})X_{\mathrm{lt}}^*]^{-1} \\
&= [X_{\mathrm{lt}}^{*\top}[(\mathcal{V}_1\mathcal{V}_2(I + \mathcal{V}_1) - \mathcal{V}_1^2\mathcal{V}_2)(I + \mathcal{V}_1)^{-1}]X_{\mathrm{lt}}^*]^{-1} \\
&= [X_{\mathrm{lt}}^{*\top}(\mathcal{V}_1\mathcal{V}_2(I + \mathcal{V}_1)^{-1})X_{\mathrm{lt}}^*]^{-1}.
\end{aligned}
$$

*All factors in $\mathcal{P}\backslash\{Y\}$ are present in the logistic regression, or no collapsing of cells.* Assume cell counts are not collapsed (by choice or when q = 1), so that $n_{\mathrm{lt}} = n_{\mathrm{ll}}/2$ and $X_{\mathrm{lt}}^* = X_{\mathrm{lt}}$. We now use the standard result (e.g. [11, (p. 200)]) that, asymptotically, the Binomial distribution Bin $(t_i, (\exp(X_{\mathrm{lt}(i)}(T_r\boldsymbol{\lambda}_r)))/(1 + \exp(X_{\mathrm{lt}(i)}(T_r\boldsymbol{\lambda}_r))))$ of a data point $t_i y_i$, $i = 1, \ldots, n_{\mathrm{lt}}$, can be approximated by Poisson $(t_i(\exp(X_{\mathrm{lt}(i)}(T_r\boldsymbol{\lambda}_r)))/(1 + \exp(X_{\mathrm{lt}(i)}(T_r\boldsymbol{\lambda}_r))))$. Considering the Poisson log-linear model, the Binomial observation $t_i - t_i \times y_i$ follows the Poisson distribution:

$$\mathrm{Poisson}\,(\exp(X_{\mathrm{ll-lt}(i)}\hat{\boldsymbol{\lambda}}_{\mathrm{ll-lt}})).$$

Therefore, approximately,

$$t_i \frac{1}{1 + \exp(X_{\mathrm{lt}(i)}(T_r\hat{\boldsymbol{\lambda}}_r))} \simeq \exp(X_{\mathrm{ll-lt}(i)}\hat{\boldsymbol{\lambda}}_{\mathrm{ll-lt}}).$$

In matrix notation, we can now write that, asymptotically,

$$
\begin{aligned}
\mathrm{Var}(T_r\hat{\boldsymbol{\lambda}}_r) &= T_r(\mathrm{Var}(\hat{\boldsymbol{\lambda}}_r))T_r^{\top} \\
&= (\, I \quad 0\,)(\mathrm{Var}(\hat{\boldsymbol{\lambda}}_r))\begin{pmatrix} I \\ 0 \end{pmatrix} \\
&= (X_{\mathrm{lt}}^{\top}\mathcal{V}_{\mathrm{logistic}}X_{\mathrm{lt}})^{-1},
\end{aligned}
$$

where $\mathcal{V}_{\mathrm{logistic}}$ has diagonal elements $t_i\exp\{X_{\mathrm{lt}(i)}\hat{\boldsymbol{\beta}}\}\exp\{1 + X_{\mathrm{lt}(i)}\hat{\boldsymbol{\beta}}\}^{-2}$, $i = 1, \ldots, n_{\mathrm{lt}}$. $(X_{\mathrm{lt}}^{\top}\mathcal{V}_{\mathrm{logistic}}X_{\mathrm{lt}})^{-1}$ is, asymptotically, the variance of $\hat{\boldsymbol{\beta}}$ when the logistic regression is fitted directly, and this completes the proof when no collapsing of cell counts takes place.

*Factors not present in the logistic regression, with collapsing of cells.* When one chooses to collapse the counts in the contingency table cells that are only discriminated by the obsolete variables $x_{.2}, \ldots, x_{.q}$,

$$H = [X_{\mathrm{lt}}^{\top}(\mathcal{V}_{1,\mathrm{reduced}}(I + \mathcal{V}_{1,\mathrm{reduced}})^{-1})][\mathcal{V}_{2,1} + \mathcal{V}_{2,2} + \cdots + \mathcal{V}_{2,(j_1-1)\times j_2 \times \cdots \times j_q}]X_{\mathrm{lt}}]^{-1},$$

where $\mathcal{V}_{1,\mathrm{reduced}}$ denotes a diagonal matrix with non-zero elements $\exp(X_{\mathrm{lt}(i)}(T_r\hat{\boldsymbol{\lambda}}_r))$, $i = 1, \ldots, n_{\mathrm{lt}}$. $\mathcal{V}_{2,k}$, $k = 1, \ldots, J_2 \times \cdots \times J_q$, denotes a diagonal matrix with elements $\exp(X_{\mathrm{ll-lt}(n_{\mathrm{lt}}(k-1)+i)}\hat{\boldsymbol{\lambda}}_{\mathrm{ll-lt}})$. Similar to the previous case, we use the standard result that, asymptotically, the Binomial distribution Bin $(t_i, (\exp(X_{\mathrm{lt}(i)}^*(T_r\boldsymbol{\lambda}_r)))/(1 + \exp(X_{\mathrm{lt}(i)}^*(T_r\boldsymbol{\lambda}_r))))$ of a data point $t_i y_i$, $i = 1, \ldots, n_{\mathrm{lt}}$, can be approximated by Poisson $(t_i(\exp(X_{\mathrm{lt}(i)}^*(T_r\boldsymbol{\lambda}_r)))/(1 + \exp(X_{\mathrm{lt}(i)}^*(T_r\boldsymbol{\lambda}_r))))$. When cell counts are collapsed, the Binomial observation $t_i - t_i \times y_i$ is formed by adding $J_2 \times \cdots \times J_q$ independent Poisson cell counts. Considering

the Poisson log-linear model, $t_i - t_i y_i$ follows the Poisson distribution:

$$\text{Poisson}(\exp(X_{\text{ll}-\text{lt}(i)}\hat{\boldsymbol{\lambda}}_{\text{ll}-\text{lt}}) + \cdots + \exp(X_{\text{ll}-\text{lt}(n_{\text{lt}}(J_2 \times \cdots \times J_q - 1)+i)}\hat{\boldsymbol{\lambda}}_{\text{ll}-\text{lt}})).$$

Therefore, approximately

$$t_i \frac{1}{1 + \exp(X_{\text{lt}(i)}(\boldsymbol{T}_r\hat{\boldsymbol{\lambda}}_r))}$$

$$\simeq \exp(X_{\text{ll}-\text{lt}(i)}\hat{\boldsymbol{\lambda}}_{\text{ll}-\text{lt}}) + \cdots + \exp(X_{\text{ll}-\text{lt}(n_{\text{lt}}(J_2 \times \cdots \times J_q - 1)+i)}\hat{\boldsymbol{\lambda}}_{\text{ll}-\text{lt}}).$$

In matrix notation, we can now write that, asymptotically

$$\begin{aligned}
\text{Var}(\boldsymbol{T}_r\hat{\boldsymbol{\lambda}}_r) &= \boldsymbol{T}_r(\text{Var}(\hat{\boldsymbol{\lambda}}_r))\boldsymbol{T}_r^{\top} \\
&= (\boldsymbol{I} \quad \boldsymbol{0})(\text{Var}(\hat{\boldsymbol{\lambda}}_r))\begin{pmatrix} \boldsymbol{I} \\ \boldsymbol{0} \end{pmatrix} \\
&\simeq [X_{\text{lt}}^{\top}(t\mathcal{V}_{1,\text{reduced}}(\boldsymbol{I} + \mathcal{V}_{1,\text{reduced}})^{-2})X_{\text{lt}}]^{-1} \\
&= (X_{\text{lt}}^{\top}\mathcal{V}_{\text{logistic}}X_{\text{lt}})^{-1},
\end{aligned}$$

where $t$ is a diagonal matrix with diagonal elements the number of trials $t_i$, and $\mathcal{V}_{\text{logistic}}$ has diagonal elements $t_i\exp\{X_{\text{lt}(i)}\hat{\boldsymbol{\beta}}\}\exp\{1 + X_{\text{lt}(i)}\hat{\boldsymbol{\beta}}\}^{-2}$, $i = 1, \ldots, n_{\text{lt}}$. $(X_{\text{lt}}^{\top}\mathcal{V}_{\text{logistic}}X_{\text{lt}})^{-1}$ is, asymptotically, the variance of $\hat{\boldsymbol{\beta}}$ when the logistic regression is fitted directly, and this completes the proof. ∎

*Proof of Theorem 3.3.* Assume that no cell observations are collapsed when one or more factors in $\mathcal{P}\backslash\{Y\}$ are not present in the logistic regression. From (2.2),

$$D(\hat{\boldsymbol{p}}, \boldsymbol{y})$$

$$= 2\sum_{j_2,\ldots,j_P} n_{1,j_2,\ldots,j_P}\log\left(\frac{n_{1,j_2,\ldots,j_P}}{n_{0,j_2,\ldots,j_P} + n_{1,j_2,\ldots,j_P}} \times \left(\frac{\exp(X_{\text{lt}(j_2,\ldots,j_P)}\hat{\boldsymbol{\beta}})}{1 + \exp(X_{\text{lt}(j_2,\ldots,j_P)}\hat{\boldsymbol{\beta}})}\right)^{-1}\right)$$

$$+ 2\sum_{j_2,\ldots,j_P} n_{0,j_2,\ldots,j_P}\log\left(\frac{n_{0,j_2,\ldots,j_P}}{n_{0,j_2,\ldots,j_P} + n_{1,j_2,\ldots,j_P}} \times \left(\frac{1}{1 + \exp(X_{\text{lt}(j_2,\ldots,j_P)}\hat{\boldsymbol{\beta}})}\right)^{-1}\right).$$

This, in turn, is equal to

$$2\sum_{j_2,\ldots,j_P} n_{1,j_2,\ldots,j_P}\log(n_{1,j_2,\ldots,j_P}) + 2\sum_{j_2,\ldots,j_P} n_{0,j_2,\ldots,j_P}\log(n_{0,j_2,\ldots,j_P}) \tag{A2}$$

$$- 2\sum_{j_2,\ldots,j_P} n_{1,j_2,\ldots,j_P}\log(\exp(X_{\text{lt}(j_2,\ldots,j_P)}\hat{\boldsymbol{\beta}})) \tag{A3}$$

and

$$- 2\sum_{j_2,\ldots,j_P} (n_{0,j_2,\ldots,j_P} + n_{1,j_2,\ldots,j_P})\log\left(\frac{n_{0,j_2,\ldots,j_P} + n_{1,j_2,\ldots,j_P}}{1 + \exp(X_{\text{lt}(j_2,\ldots,j_P)}\hat{\boldsymbol{\beta}})}\right). \tag{A4}$$

For the log-linear model, from (2.1),

$$\begin{aligned}
D(\hat{\boldsymbol{\mu}}, \boldsymbol{n}) &= 2\sum_{i=1}^{n_{\text{ll}}} n_i\log\left(\frac{n_i}{\hat{\mu}_i}\right) \\
&= 2\sum_{j_2,\ldots,j_P} n_{0,j_2,\ldots,j_P}\log\left(\frac{n_{0,j_2,\ldots,j_P}}{\hat{\mu}_{0,j_2,\ldots,j_P}}\right) + 2\sum_{j_2,\ldots,j_P} n_{1,j_2,\ldots,j_P}\log\left(\frac{n_{1,j_2,\ldots,j_P}}{\hat{\mu}_{1,j_2,\ldots,j_P}}\right) \\
&= 2\sum_{j_2,\ldots,j_P} n_{0,j_2,\ldots,j_P}\log\left(\frac{n_{0,j_2,\ldots,j_P}}{\exp(X_{\text{ll}(0,j_2,\ldots,j_P)}\hat{\boldsymbol{\lambda}})}\right) \\
&\quad + 2\sum_{j_2,\ldots,j_P} n_{1,j_2,\ldots,j_P}\log\left(\frac{n_{1,j_2,\ldots,j_P}}{\exp(X_{\text{ll}(1,j_2,\ldots,j_P)}\hat{\boldsymbol{\lambda}})}\right)
\end{aligned}$$

This, in turn, is equal to

$$2 \sum_{j_2,\ldots,j_P} n_{0,j_2,\ldots,j_P} \log(n_{0,j_2,\ldots,j_P}) + 2 \sum_{j_2,\ldots,j_P} n_{1,j_2,\ldots,j_P} \log(n_{1,j_2,\ldots,j_P}) \tag{A5}$$

$$- 2 \sum_{j_2,\ldots,j_P} n_{1,j_2,\ldots,j_P} (X_{\mathrm{rll}(1,j_2,\ldots,j_P)}[1:n_\beta]\hat{\boldsymbol{\lambda}}_r[1:n_\beta]) \tag{A6}$$

$$- 2 \sum_{j_2,\ldots,j_P} n_{0,j_2,\ldots,j_P} X_{\mathrm{rll}(0,j_2,\ldots,j_P)}[n_\beta+1:n_\lambda]\hat{\boldsymbol{\lambda}}_r[n_\beta+1:n_\lambda] \tag{A7}$$

and

$$- 2 \sum_{j_2,\ldots,j_P} n_{1,j_2,\ldots,j_P} X_{\mathrm{rll}(1,j_2,\ldots,j_P)}[n_\beta+1:n_\lambda]\hat{\boldsymbol{\lambda}}_r[n_\beta+1:n_\lambda]. \tag{A8}$$

Now, (A2)=(A5) by inspection. Furthermore, from theorem 3.2, $\hat{\boldsymbol{\beta}} = \hat{\boldsymbol{\lambda}}_r[1:n_\beta]$. As,

$$X_{\mathrm{rll}(1,j_2,\ldots,j_P)}[1:n_\beta]\hat{\boldsymbol{\lambda}}_r[1:n_\beta] = X_{\mathrm{lt}(j_2,\ldots,j_P)}\hat{\boldsymbol{\beta}},$$

we have that (A3)=(A6). Finally, from Lemma 3.1,

$$n_{0,j_2,\ldots,j_P} + n_{1,j_2,\ldots,j_P} = \hat{\mu}_{0,j_2,\ldots,j_P} + \hat{\mu}_{1,j_2,\ldots,j_P}.$$

Also,

$$1 + \exp(X_{\mathrm{lt}(j_2,\ldots,j_P)}\hat{\boldsymbol{\beta}}) = \frac{1}{\hat{p}_{0,j_2,\ldots,j_P}}.$$

Then,

$$(\mathrm{A}\,4) = -2 \sum_{j_2,\ldots,j_P} n_{1,j_2,\ldots,j_P} \log\left(\frac{\hat{\mu}_{0,j_2,\ldots,j_P} + \hat{\mu}_{1,j_2,\ldots,j_P}}{1/\hat{p}_{0,j_2,\ldots,j_P}}\right)$$

$$- 2 \sum_{j_2,\ldots,j_P} n_{0,j_2,\ldots,j_P} \log\left(\frac{\hat{\mu}_{0,j_2,\ldots,j_P} + \hat{\mu}_{1,j_2,\ldots,j_P}}{1/\hat{p}_{0,j_2,\ldots,j_P}}\right)$$

$$= -2 \sum_{j_2,\ldots,j_P} n_{1,j_2,\ldots,j_P} \log(\hat{\mu}_{0,j_2,\ldots,j_P}) - 2 \sum_{j_2,\ldots,j_P} n_{0,j_2,\ldots,j_P} \log(\hat{\mu}_{0,j_2,\ldots,j_P}) = (\mathrm{A7}) + (\mathrm{A8}).$$

This completes the proof of theorem 3.4. ∎

*Data analysed in §4.* The dataset for log-linear model (M2) is given by vector

$$\boldsymbol{n} = (44, 40, 112, 67, 129, 145, 12, 23, 35, 12, 80, 33, 109, 67, 7, 9, 23, 32, 70, 66, 50,$$
$$80, 7, 13, 24, 25, 73, 57, 51, 63, 7, 16, 5, 7, 21, 9, 9, 17, 1, 4, 4, 3, 11, 8, 14, 17, 5, 2, 7,$$
$$3, 14, 14, 9, 16, 2, 3, 4, 0, 13, 11, 5, 14, 4, 4).$$

The dataset for the logistic regression (M3) is

$$\boldsymbol{t} = (84, 179, 274, 35, 47, 113, 176, 16, 55, 136, 130, 20, 49, 130, 114, 23, 12, 30, 26,$$
$$5, 7, 19, 31, 7, 10, 28, 25, 5, 4, 24, 19, 8),$$

$$\boldsymbol{y} = \left(\frac{40}{84}, \frac{67}{179}, \frac{145}{274}, \frac{23}{35}, \frac{12}{47}, \frac{33}{113}, \frac{67}{176}, \frac{9}{16}, \frac{32}{55}, \frac{66}{136},\right.$$

$$\frac{80}{130}, \frac{13}{20}, \frac{25}{49}, \frac{57}{130}, \frac{63}{114}, \frac{16}{23}, \frac{7}{12}, \frac{9}{30}, \frac{17}{26}, \frac{4}{5}, \frac{3}{7}, \frac{8}{19},$$

$$\left.\frac{17}{31}, \frac{2}{7}, \frac{3}{10}, \frac{14}{28}, \frac{16}{25}, \frac{3}{5}, \frac{0}{4}, \frac{11}{24}, \frac{14}{19}, \frac{4}{8}\right).$$

The dataset for (M4) is,

$$\boldsymbol{t} = (305, 340, 186, 230, 229, 180, 207, 164)$$

and

$$\boldsymbol{y} = \left(\frac{123}{305}, \frac{189}{340}, \frac{56}{186}, \frac{95}{230}, \frac{115}{229}, \frac{112}{180}, \frac{93}{207}, \frac{97}{164}\right).$$

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
