## [Reviewer comments · Royal Society Open Science]

Review History

RSOS-191483.R0 (Original submission)

Review form: Reviewer 1

Is the manuscript scientifically sound in its present form?

Yes

Are the interpretations and conclusions justified by the results?

Yes

Is the language acceptable?

Yes

Do you have any ethical concerns with this paper?

No

Have you any concerns about statistical analyses in this paper?

No

Recommendation?

Accept as is

Comments to the Author(s)

page 3, line 45 "models" should be "model"

writing contingency table data as vectors is kind of unsatisfying even though it is obviously the form you need it in.

Review form: Reviewer 2

Is the manuscript scientifically sound in its present form?

Yes

Are the interpretations and conclusions justified by the results?

Yes

Is the language acceptable?

Yes

Do you have any ethical concerns with this paper?

No

Have you any concerns about statistical analyses in this paper?

No

Recommendation?

Major revision is needed (please make suggestions in comments)

Comments to the Author(s)

See the attached pdf (Appendix A).

Decision letter (RSOS-191483.R0)

11-Nov-2019

Dear Dr Papathomas,

On behalf of the Editors, I am pleased to inform you that your Manuscript RSOS-191483 entitled "On the correspondence of deviances and maximum likelihood and interval estimates from log-linear to logistic regression modelling" has been accepted for publication in Royal Society Open Science subject to minor revision in accordance with the referee suggestions. Please find the referees' comments at the end of this email.

The reviewers and handling editors have recommended publication, but also suggest some minor revisions to your manuscript. Therefore, I invite you to respond to the comments and revise your manuscript.

- Ethics statement

- Data accessibility

<http://datadryad.org/submit?journalID=RSOS&manu=RSOS-191483>

- Competing interests

- Authors' contributions

- Acknowledgements

- Funding statement

Please ensure you have prepared your revision in accordance with the guidance at <https://royalsociety.org/journals/authors/author-guidelines/> -- please note that we cannot publish your manuscript without the end statements. We have included a screenshot example of

the end statements for reference. If you feel that a given heading is not relevant to your paper, please nevertheless include the heading and explicitly state that it is not relevant to your work.

Because the schedule for publication is very tight, it is a condition of publication that you submit the revised version of your manuscript before 20-Nov-2019. Please note that the revision deadline will expire at 00.00am on this date. If you do not think you will be able to meet this date please let me know immediately.

Please note that Royal Society Open Science charge article processing charges for all new submissions that are accepted for publication. Charges will also apply to papers transferred to Royal Society Open Science from other Royal Society Publishing journals, as well as papers

submitted as part of our collaboration with the Royal Society of Chemistry (<https://royalsocietypublishing.org/rsos/chemistry>).

If your manuscript is newly submitted and subsequently accepted for publication, you will be asked to pay the article processing charge, unless you request a waiver and this is approved by Royal Society Publishing. You can find out more about the charges at <https://royalsocietypublishing.org/rsos/charges>. Should you have any queries, please contact openscience@royalsociety.org.

Kind regards,

on behalf of Professor Ruth King (Associate Editor) and Mark Chaplain (Subject Editor)
openscience@royalsociety.org

Associate Editor Comments to Author (Professor Ruth King):

The paper has been reviewed by two referees and myself. The paper is very well and concisely written. Both reviewers (and myself) found the paper interesting - though the practical implications were perhaps less clear (note - I do not see this as a significant issue - though adding some comment on this may be useful). The main issue that should be addressed relates to providing a clear explanation of why the results presented in the paper are novel and not a special case of the general previously published results of the relationship between log-linear and logistic regression models for contingency table data. This will help clarify the novelty of the paper to the reader. Referee 2 has also made a few suggestions to improve the paper, most notably adding in details of the general case for those readers who may be less familiar with these results.

Reviewer comments to Author:

Reviewer: 1

Comments to the Author(s)
page 3, line 45 "models" should be "model"

writing contingency table data as vectors is kind of unsatisfying even though it is obviously the form you need it in.

Reviewer: 2
Comments to the Author(s)

See the attached pdf file.

Author's Response to Decision Letter for (RSOS-191483.R0)

See Appendix B.

Decision letter (RSOS-191483.R1)

28-Nov-2019

Dear Dr Papathomas,

It is a pleasure to accept your manuscript entitled "On the correspondence of deviances and maximum likelihood and interval estimates from log-linear to logistic regression modelling" in its current form for publication in Royal Society Open Science.

on behalf of Professor Ruth King (Associate Editor) and Mark Chaplain (Subject Editor)
openscience@royalsociety.org

Appendix A

On the correspondence of deviances and maximum likelihood and interval estimates from log-linear to logistic regression modelling

Submitted to *Royal Society Open Science*

Manuscript RSOS-191483

General comments

The paper is interesting and easy to follow. The main contribution of the paper is that it generalizes a well known result for the connection between then a log-linear model and a logistic regression model. The main generalizations are

1. For the case when some factors are not included in the logistic regression model
2. When we collapse some of the variables in the log-linear model and the contingency table
3. About the equality of the deviance measures.

In the past (i.e. 90s) this correspondence was important since you could easily fit logistic and/or multinomial regression models through this linkage. Nevertheless, this connection seems to be only of academic curiosity and understanding of the theory and modelling structure and interpretation of these two models. Although this paper can serve as a useful reference for research statisticians and possibly for teaching categorical data and/or GLM, it is not clear to me where these findings are useful nowadays. Moreover, they intuitively seem to me as special cases of the general association between the models. My final concern is whether and how this paper fits in the interest of the reader of *RSOS* which seems to be a journal of more general scientific findings.

Main comments

My main comments on the paper are the following

1. **General comment:** I would expect to present the general case of the association between the multinomial logistic regression model and the log-linear model
2. **General comment:** Motivate how and why this work and the associated results are of interest or useful nowadays.

3. **General comment:** Motivate how and why this article is useful or insightful for the general reader of RSOS
4. **General Comment:** Explain why the results you present are NEW and cannot be considered as a special case of the general connection between the log-linear and the logistic models.

Specific and minor comments

1. **Page 6, Theorem 3.3:** Explain what are n_{li} and n_{ll} and why the corresponding equality is important to hold or how it is derived.
2. **Page 8, Tables 1 & 2:** These tables are messy. Please merge them in one with three rows (one for every model) with the coefficients followed by the standard errors in brackets (numbers to be compared should be on the same column). Use the same number of decimals in all numbers (3 or 4). Also include the deviance measures for all three models under comparison.
3. **General Comment:** In the paper you refer to merging the factors of a contingency table which in my opinion is confusing since in contingency tables merging usually refers to the categories/levels of a factor. I would recommend using the term collapsing or marginalizing over since you obtain the models from another marginal table.

Appendix B

19 November 2019
Dr Michail Papathomas
Senior Lecturer in Statistics
School of Mathematics and Statistics
University of St Andrews
The Observatory, Buchanan Gardens
St Andrews, KY16 9LZ, UK
UK Tel: +44 (0) 1334 461818, +44 7766891768
Email: M.Papathomas@st-andrews.ac.uk

Professor Jeremy Sanders CBE FRS (Editor in Chief)
Royal Society Open Science

Dear Professor Sanders,

Re: Revision of manuscript ‘On the correspondence of deviances and maximum likelihood and interval estimates from log-linear to logistic regression modelling’.

Please consider the revised version of manuscript RSOS-191483 with title ‘On the correspondence of deviances and maximum likelihood and interval estimates from log-linear to logistic regression modelling’.

We believe we have addressed the comments made by the Associate Editor and the two Reviewers. Please see the attached detailed response over the next five pages, regarding all the valuable comments that improved this manuscript.

Our revision did not alter the length of the manuscript.

Yours sincerely,
Wei Jing and Michail Papathomas

RSOS-191483 – Response to the Associate Editor

The paper is very well and concisely written. Both reviewers (and myself) found the paper interesting - though the practical implications were perhaps less clear (note - I do not see this as a significant issue - though adding some comment on this may be useful). The main issue that should be addressed relates to providing a clear explanation of why the results presented in the paper are novel and not a special case of the general previously published results of the relationship between log-linear and logistic regression models for contingency table data. This will help clarify the novelty of the paper to the reader. Referee 2 has also made a few suggestions to improve the paper, most notably adding in details of the general case for those readers who may be less familiar with these results.

Thank you for your positive remarks and suggestions. We have prepared a revised manuscript that addresses your suggestions and the comments by the two reviewers.

We have now added a sentence in the last paragraph of the Introduction, pointing to the Discussion section where possible practical implications of our work are considered.

Furthermore, we provide a clearer explanation on the novelty of our results, and introduce material on the general case of the correspondence we study. We hope we have responded appropriately. Please see our response to all the comments by the two Reviewers in the four pages below.

RSOS-191483 – Response to Reviewer 1

page 3, line 45 "models" should be "model"

This is now done. Thank you.

writing contingency table data as vectors is kind of unsatisfying even though it is obviously the form you need it in.

We agree there are better, more intuitive ways for presenting contingency table data. However, as we consider the log-linear model within the Generalized Linear Modelling framework, drawing parallels with Logistic Regression, this form best suits the problem at hand.

Thank you for your comments.

RSOS-191483 – Response to Reviewer 2

The paper is interesting and easy to follow. The main contribution of the paper is...

Thank you for your suggestions and positive comments that improved this manuscript.

1. General comment: I would expect to present the general case of the association between the multinomial logistic regression model and the log-linear model

We now describe the more general association between multinomial logistic regression and the log-linear model in the fourth and fifth paragraphs of the Introduction. We do not provide a very detailed account of this association for two reasons. First, the focus of this manuscript is on binomial logistic regression. Second, to avoid a notable increase in the size of the manuscript.

2. General comment: Motivate how and why this work and the associated results are of interest or useful nowadays

We believe the manuscript results are of interest and useful for the following reasons. First, as we also discuss in our response to comment 3 below, such results are quite important for gaining insight on the nature and properties of two fundamental and widely used modelling approaches. For example, understanding the effect of collapsing cell counts on the deviance of the fitted logistic regression, whilst this collapsing operation does not alter point and interval estimates for the model parameters, is quite instructive.

Second, scientific developments often only concern one of the two modelling frameworks, and are not readily available for the other. One of the two examples we mention in the Discussion section is Papathomas and Richardson [2016], where the utility of employing variable selection within clustering to assist log-linear modelling is investigated, without examining (multinomial) logistic regression models. Among other examples not discussed in the manuscript, is model checking in log-linear models using Φ -divergences and MLEs (Cressie and Pardo; 2002), and recent developments in the identifiability and estimability of log-linear model parameters; see Chan et al. (2019) and Far et al. (2019).

Cressie, N. and Pardo, L. (2002). Model checking in loglinear models using ϕ -divergences and MLEs. *J. Statist. Plann. Inference* **103**, 437-453.

Chan, L., Silverman, B.W. and Vincent, K. (2019) Multiple systems estimation for sparse capture data: inferential challenges when there are non-overlapping lists. *Preprint. arXiv:1902.05156*.
Sharifi Far S, Papatomas M. and King R. (2019): Parameter redundancy and the existence of MLE in Log-linear Models. *Statistica Sinica*. doi: 10.5705/ss.202018.0100

Finally, we believe results on the equality between the deviances can be useful in terms of model comparison. Intuitively, once someone identifies a ‘best’ (for some criterion) log-linear model for a set of categorical factors, this should provide a very good indication on what main effects and interactions should be included in a good (if not ‘best’) logistic regression for any of the binary factors in that set. This manuscript may provide the motivation for future research to substantiate this intuitive argument, and possibly extend it for the case of multinomial logistic regression. We have not included this argument in the Discussion section, as at present we cannot support it with a completed study.

3. General comment: Motivate how and why this article is useful or insightful for the general reader of RSOS

This manuscript concerns the correspondence between two of the most popular modelling approaches utilised by both statisticians and practitioners. For example, log-linear modelling is currently the main approach for estimating the size of hidden populations such as victims of modern slavery; see Silverman (2019).

Silverman, B.W. (2019). Multiple-systems analysis for the quantification of modern slavery: classical and Bayesian approaches. Read paper at the November 2019 Royal Statistical Society meeting.

Examples of the popularity of logistic regression modelling are numerous, for instance in the field of epidemiology.

Our manuscript contains mathematical proofs of considerable complexity that extend results on the equivalence between the two modelling approaches. We believe early results, as well as the new results shown in this manuscript, are quite important for gaining insight on the nature and properties of these two fundamental modelling approaches. For example, understanding the effect of collapsing cell counts on the deviance of the fitted logistic regression model, whilst this collapsing operation does not actually alter point and interval estimates for the model parameters, is quite instructive.

In fact, we hope that the new results in this manuscript will help bring to the attention of statisticians and practitioners, as well as teachers of statistics, the standard results on the equivalence between these two modelling approaches, and possibly other earlier results such as Bishop (1971).

Bishop, Y.M.M. (1971). Effects of collapsing multidimensional contingency tables. *Biometrics*. 27, 545-562.

Please see also our response to comment 2 above, on the usefulness of the manuscript results.

4. General Comment: Explain why the results you present are NEW and cannot be considered as a special case of the general connection between the log-linear and the logistic models.

The first new result is that the correspondence (in terms of point and interval parameter estimates) between the log-linear model and the corresponding logistic regression holds even when contingency table factors are not present in the corresponding logistic regression and some of the contingency table cells are collapsed together. This case is not considered in the textbooks that study or discuss this equivalence, namely Christensen (1997) and Agresti (2002). We now clarify this

on page 3 in the Introduction section. It is necessary to provide additional mathematical justification for this result, and this is what we offer in this manuscript. We are not aware of any published work where this result is proven, also after discussions we had with Professor Christensen.

The second new result is on the equality of deviances for the log-linear model and corresponding logistic regression, given a certain condition. As we discuss within the manuscript (towards the end of Section 1), Christensen (1997, p. 371) refers to this equality, but proof is only given for a restricted case of a simple logistic regression with two parameters. Specifically, it is shown that the likelihood ratio test statistic (LRTS) for the log-linear model equals the LRTS for the logistic regression, utilizing the invariance of the MLE and the properties of the product-binomial sampling scheme (Christensen, 1997, Section 2.6). Christensen (1997, p. 365) also shows that applying the logistic regression to a contingency table implies that the sampling scheme of the contingency table is product-binomial instead of multinomial, implying that the test statistic is identical for the two models. As the relevant mathematical results shown in Christensen (1997, Section 2.6) are based on a simple logistic regression with two parameters, a general mathematical proof is required, and this is what we provide in this manuscript. We now clarify this in the 5th paragraph on page 3. Within the manuscript we describe the work by Christensen in slightly less detail than in our response here, as the reader can refer to the relevant textbook. When we discussed our point of view with Professor Christensen, there was no indication that our understanding of this work was incorrect.

Page 6, Theorem 3.3: Explain what are n_{lt} and n_{ll} and why the corresponding equality is important to hold or how it is derived.

The n_{lt} and n_{ll} quantities are initially defined in Section 2. We now also state what they represent just before Theorem 3.3. In addition, we explain intuitively why the corresponding equality is important. Specifically, the few lines before stating Theorem 3.3 have now changed to: ‘Theorem 3.3 postulates that $n_{lt}=n_{ll}/2$, i.e. the number of proportions fitted by the logistic regression should be half the number of cell counts in the contingency table. This happens either because all factors in $\mathcal{P} \setminus \{ Y \}$ are present in the logistic regression, or because counts in cells with the same cross-classification considering $x_{.q+1}, \dots, x_{.P}$ are not collapsed. This is important for observing equal deviances for the log-linear model and the corresponding logistic regression. Intuitively, when $n_{lt}=n_{ll}/2$, the number of observations fitted by the logistic regression is in direct correspondence with the number of observations fitted by the log-linear model. When $n_{lt}<n_{ll}/2$, a logistic regression model with the same number of parameters fits a smaller number of observations, something that naturally results in a smaller deviance compared to the deviance observed when the contingency table is not collapsed. This is illustrated in Section 4 with the analysis of a real data set.

Page 8, Tables 1 & 2: These tables are messy. Please merge them in one with three rows (one for every model) with the coefficients followed by the standard errors in brackets (numbers to be compared should be on the same column). Use the same number of decimals in all numbers (3 or 4). Also include the deviance measures for all three models under comparison.

Done. Thank you for pointing this out. We have merged the two Tables in accordance with your suggestions. The deviances are presented within the new Table.

General Comment: In the paper you refer to merging the factors of a contingency table which in my opinion is confusing since in contingency tables merging usually refers to the categories/levels of a factor. I would recommend using the term collapsing or marginalizing over since you obtain the models from another marginal table.

We agree with this comment. We decided that the term ‘collapsing’ is preferable, as marginalising is associated with distributional operations too, and this may cause some confusion. The term ‘collapsing’ or ‘collapsed together’ is now used instead of the term ‘merging’ throughout the manuscript.

Thank you for your comments which helped us to improve the manuscript.